psychology

momentary affect, goal-relevance, affective integration, affective dynamics, attention, affective experience

**Author for correspondence:**
Erkin Asutay
e-mail: erkin.asutay@liu.se

# The goal-relevance of affective stimuli is dynamically represented in affective experience

Erkin Asutay[1] and Daniel Västfjäll[1,2]

[1]Department of Behavioral Sciences and Learning, Linköping University, Sweden
[2]Decision Research, Eugene, OR, USA

 EA, 0000-0002-4257-6732

Affect is a continuous and temporally dependent process that represents an individual's ongoing relationship with its environment. However, there is a lack of evidence on how factors defining the dynamic sensory environment modulate changes in momentary affective experience. Here, we show that goal-dependent relevance of stimuli is a key factor shaping momentary affect in a dynamic context. Participants ($N = 83$) viewed sequentially presented images and reported their momentary affective experience after every fourth stimulus. Relevance was manipulated through an attentional task that rendered each image either task-relevant or task-irrelevant. Computational models were fitted to trial-by-trial affective responses to capture the key dynamic parameters explaining momentary affective experience. The findings from statistical analyses and computational models showed that momentary affective experience was shaped by the temporal integration of the affective impact of recently encountered stimuli, and that task-relevant stimuli, independent of stimulus affect, prompted larger changes in experienced pleasantness compared with task-irrelevant stimuli. These findings clearly show that dynamics of affective experience reflect goal-relevance of stimuli in our surroundings.

## 1. Introduction

Humans navigate complex environments, in which we receive a continuous stream of evocative stimuli that induces changes in our momentary affective state. However, the critical question of how the affective impact prompted by a stream of sensory information is dynamically represented in momentary affect remains largely unanswered. It is widely hypothesized that affect is related to sensory changes within the body due to fluctuations in physiological systems [1,2]. Our brains

continually represent these physiological adaptations in response to changing environmental demands to keep us alive [3]. Hence, affective feelings infuse every waking moment of our lives [4], and they represent our ongoing relationship with the environment [5,6]. This means that affect is a temporally dependent and continuous process, and an individual's affective state at a given time reflects the changes in environmental circumstances and the individual's affective state at earlier time points [7]. In fact, the dynamic models of affect suggest that fluctuations of affective experience are influenced by several factors such as expectations, context and previous affective state (e.g. [8]). Recent empirical findings show that momentary affective experience is shaped by temporal integration of the affective impact of recently encountered stimuli and individual's previous affective state [9,10]. Nevertheless, there are still unanswered questions on how different factors defining the dynamic sensory environment influence this form of affective integration. Here, we tested the hypothesis that behavioural relevance of affective stimuli is one of the determining factors shaping fluctuations in momentary affective experience.

The behavioural relevance of stimuli is a critical aspect of the sensory environment. Humans interact with their surroundings with intention and goal-directed action [11] leading us to prioritize certain stimuli over others. Thus, under a certain state (e.g. thirst) certain things (e.g. water) we may encounter have higher behavioural relevance than other stimuli (e.g. food). It is not surprising that this form of goal-dependent stimulus relevance greatly influences sensory processing [12–14]. In fact, previous investigations show that attentional selection and control occurs according to a priority map that defines the behavioural importance (including goal-relevance, selection history and physical salience) of objects in our surroundings [11,15,16]. Goal-relevance may also influence working memory and internal representations of stimuli ([17]; see also [18,19]). In addition, when manipulated through changing task goals, stimulus relevance modulates feature representations in frontoparietal attention networks [20]. Taken together, the behavioural relevance of sensory input influences information processing, attentional selection and internal representations.

Moreover, goal-relevance of affective information may be a determining factor of enhanced attentional selection and strengthened memory [21]. The abundant literature on affect and attention suggests that affective salience of stimuli and experienced emotion can influence attentional selection (e.g. [21–24]). The current research concerns the reverse relationship; that is, whether voluntary attentional selection informed by stimulus relevance influences fluctuations in momentary affective experience. Previous investigations showed that previously ignored neutral stimuli are evaluated more negatively than previously attended stimuli [25] and that previously attended visual images are evaluated as more emotionally intense in comparison with control stimuli [26]. These findings show that voluntary attentional selection can influence the evaluation of sensory stimuli. Considering also the temporally dependent nature of affect and other mental events [7], we hypothesized that voluntary attentional selection history that is informed by stimulus relevance is represented in moment-to-moment changes in affective experience. Thus, we tested whether goal-relevance of stimuli is reflected in momentary affective experience in a dynamic context.

We use empirical and computational methods to study the relationship between affective fluctuations and stimulus relevance in a dynamic context, controlling for affective salience of stimuli. Using a paradigm for affective integration from our laboratory [9,10], participants viewed sequentially presented images and reported their momentary affective experience (How do you feel right now?) after every fourth image using two visual analogue scales: valence (pleasant to unpleasant) and arousal (sleepiness to high activation). To manipulate stimulus relevance, we incorporated an attentional task into this basic paradigm. Participants identified a target presented at a random location on the images as accurately and quickly as possible. A relevancy cue was presented prior to each image indicating whether the next image was task-relevant or not (figure 1a). Importantly, participants were not told to actively suppress a response during task-irrelevant stimuli; instead, they were explicitly instructed that identified targets presented with irrelevant images will not be counted. Our aim was to manipulate voluntary attentional control by rendering a subset of stimuli as irrelevant for the current task goals and to investigate the effect of stimulus relevance on trial-by-trial changes in self-reported affect.

In addition, we adopt a computational modelling approach to capture the key dynamic variables of interest shaping affective experience. Arguably, computational models aimed to parametrize how affect changes as a function of the sensory information flow will greatly contribute to our understanding of affective experience as a temporally dependent mental process. Therefore, we fitted two computational models to trial-by-trial affective experience. One of the models conceptualizes momentary subjective affective states as the integration of the affective impact of previously encountered stimuli and events [28,29]. According to this model, ratings of subjective affective experience are generated based on an

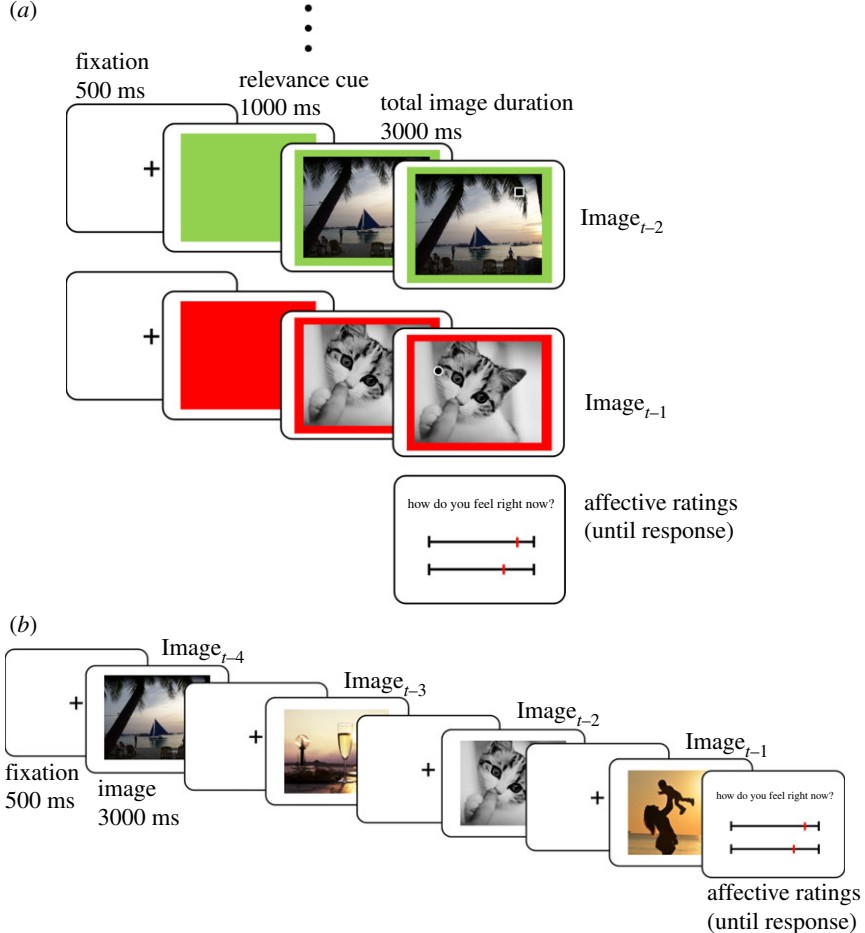

**Figure 1.** Study design. (*a*) The main task was to identify the shape of a target (a circle or a rectangle) presented on top of the images by pressing a corresponding button as quickly and accurately as possible. Prior to images, a relevancy cue was presented rendering the images as either task-relevant (green cue) or task-irrelevant (red cue). Participants were instructed that responses during task-irrelevant images do not count. After every fourth image, participants reported their momentary affective experience. (Image$_{t-1}$: the last image viewed before an affect rating). (*b*) Trial structure of the control block. Participants viewed sequentially presented images and reported their momentary affective experience after every fourth image (Image$_{t-1}$ to Image$_{t-4}$: the four stimuli prior to an affect rating). Images are from the OASIS database [27].

exponential decay (ED) of the influences of previous events. We also formulated and tested an additional model that focuses on the momentary changes in affective experience prompted by sensory input. According to this model, the sensory input prompts a change in momentary affect proportional to the weighted difference between the input's normative tendency to induce an affective change and the individual's current affective experience. We fitted both models to self-reported valence and arousal and studied how the relevance of evocative information influences momentary affect. Based on previous research, we expected attentional selection and internal representations to be enhanced for task-relevant stimuli. Hence, we hypothesized that task-relevant stimuli, in comparison with task-irrelevant stimuli, would have a greater impact on trial-to-trial changes in affective experience.

# 2. Material and methods

## 2.1. Participants

A power analysis using G*Power (Version 3; [30]) indicated that a minimum sample of 54 is necessary to detect a within-subject effect of stimulus relevance (task-relevant versus task-irrelevant stimuli) on momentary affect with a medium effect size (*dz*) of 0.5 with a 0.95 power and $\alpha = 0.05$. Prior to the experiment, the minimum duration of the data collection was decided to be two weeks, after which

we concluded the study since we had a larger sample than required by the power analysis (89 individuals, 37 women, 50 men, two did not disclose; mean age = 24.76, s.d. = 4.98).

Individuals were recruited through a university participant pool mostly consisting of college students. They gave informed consent prior to inclusion in the experiment and were compensated after the study. The study was conducted in accordance with the ethical standards in the Declaration of Helsinki. We excluded six participants from all analyses who responded without moving the slider on either valence or arousal scales more than 80% of the time (i.e. more than 76 out of 95 affect ratings). Our final sample size thus consisted of 83 individuals (34 women, 47 men, two did not disclose; mean age = 24.64, s.d. = 4.59).

## 2.2. Materials, experimental design and procedure

The main task in the experiment was to identify a target shape presented on visual images (figure 1*a*). Prior to each image, a colour cue (a green or a red rectangle) was presented for 1000 ms to signal whether the next image was relevant (green) or not (red). During target-images, a target (a circle or a rectangle) was presented at a random location on the image. Participants' task was to identify the shape of the target via a button press as quickly and accurately as possible. The target onset was set randomly between 1500 and 2000 ms from the image onset. The image size was 625 × 500 pixels, while the target size was 50 × 50 pixels. Regardless of the condition (relevant or irrelevant and target or no-target) and response, each image stayed on the screen for 3000 ms. Any response after this was counted as a miss. Participants were explicitly instructed to inspect all the images during the task since they may be asked about the image content later in the study. Furthermore, participants were instructed that they could identify the targets presented on irrelevant images if they like; however, those responses would not be counted as correct or false. After every fourth image, participants were asked to report their momentary affective experience on two visual analogue scales: valence (pleasant to unpleasant) and arousal (sleepiness to high-activation). Here, participants were instructed to assess how they currently feel. They were told to 'look inwards' and assess how they feel at the moment [9,10]. Momentary affect ratings were standardized individually.

Participants performed the main task in four blocks of 76 images (304 images in total, 19 measurements of momentary affective experience per block). In the main task, a target occurred during 52.6% of the images (160 images). The rest were no-target images, which were distributed randomly in each block. Task-relevant and task-irrelevant images were equally distributed between blocks, but they were presented in a random order within each block with the requirement that the number of task-relevant (or task-irrelevant) stimuli did not differ among different temporal positions between two consecutive affect ratings. Moreover, participants went through a control block of 76 images, wherein the main task was removed (figure 1*b*). In this block, participants simply viewed the images (3 s per image) and reported their momentary affective experience after every fourth image on valence and arousal scales. The control block was either the first or the last block in the study, which was balanced among participants. This block of trials was included in the study since this is the form of the paradigm used in earlier studies [9,10]. Therefore, we aimed at replicating those findings. Additionally, this block of trials enabled us to study the computational modelling results when the paradigm did not contain an attentional task and was instead a passive viewing paradigm.

Visual stimuli were acquired from the OASIS database [27] complete with normative valence and arousal data (measured on 7-point scales, ranging from 1 = very negative or very low arousal to 7 = very positive or very high arousal). We first removed all neutral images (normative valence between 3.5 and 4.5). From the remaining stimuli, we selected 200 images (100 pleasant and 100 unpleasant images matched in arousal). The selected images were ensured to have various content. The final set of pleasant images included content such as babies, couples kissing, cute animals, flowers, food, nature scenery and various activities (e.g. bungee jumping, diving, rafting, rollercoasters). Unpleasant images included aggressive animals, bodily injuries, scenes of war and destruction, and crying and agitated individuals. During the study, most of the images were presented twice. However, the two presentations of an image never occurred in the same block. Images were assigned to trials for each individual randomly with the following requirements. Equal number of pleasant and unpleasant images were presented in each block, at each temporal position between two consecutive affect ratings (i.e. 1–4 images before an affect rating), and in each relevancy (relevant versus irrelevant) and target conditions (circle, rectangle and no-target). In addition, we performed a series of one-way ANOVAs to make sure that normative valence or arousal of images did not differ between different blocks, different temporal positions, or different relevancy and target conditions (all at $p > 0.25$ level).

The study was carried out in a computer laboratory. Participants were admitted to the room in groups (maximum 12 participants in a session). Each participant sat in front of a 21-inch computer screen at a comfortable distance. Partition panels were placed between the individuals to block their vision for other participants' screens. At the end of the study, participants were debriefed and compensated.

## 2.3. Data analyses and modelling

### 2.3.1. Task metrics

The task performance was analysed to ensure that participants understood and performed the task correctly. Task metrics included hits, false alarms and reactions times (RTs). We reported hit and false alarm rates instead of d-prime as an index of task performance [31], since the task yielded overall high hit rates and no false alarms, which produces d-prime scores at the very high end of the spectrum that are correlated with hit rates. In addition, we investigated the effects of normative image affect and block order on RTs using a generalized linear mixed model (GLMM) with *fitglme* function in Matlab (version R2019b). This model enabled us to analyse RTs for correctly identified targets presented with relevant images and contained fixed effects of normative valence and arousal of a given image and block order. The model also contained random intercepts and slopes at the participant level. Hence, both the intercept and the estimates of the predictors were allowed to vary across individuals.

### 2.3.2. Analysis of momentary affective experience

We analysed the relationship between the momentary affective experience and the normative tendency of the stimuli to induce an affective change, which was represented by the normative valence and arousal. Thus, normative valence and arousal of images were taken as a proxy for the normative affective impact expected from the stimuli (see also [9,10]). The analysis was carried out using *fitglme* function in Matlab. The statistical models contained fixed effects of the normative valence and arousal of the four images presented before each affect rating. The models also contained random intercepts and slopes at the participant level. Wald tests were used for *post hoc* comparisons of stimated effects for each image. Holm–Bonferroni corrections were applied to correct for multiple comparisons.

Critically, to investigate the impact of image relevance on momentary affective experience, we performed an additional GLMM analysis predicting valence and arousal reported only during the task blocks. This model contained fixed effects of image affect (normative valence or arousal) and the interaction between image affect and relevance for the four stimuli presented before an affect rating.

### 2.3.3. Computational models

Self-reported valence and arousal were fitted using two computational models that differ in their conceptualizations of momentary affect. The free parameters in both models were fitted to self-reported valence and arousal for each participant with maximum-likelihood estimation assuming a normal likelihood function. The estimation was done with the *fmincon* function in Matlab (version 2019b).

#### 2.3.3.1. Model #1 (exponential decay)

The ED model was based on a previously published and validated model that generates a momentary subjective affective state based on the integration of previously encountered information [28,29]. According to this model, subjective affective state at a given time is generated based on an ED of the influences of previous stimuli or events. Hence, at each time point, the affective impact of previously encountered events is integrated. The model has been used to study momentary happiness ratings of individuals performing a risky choice task [28]. In the current study, subjective affective ratings are two dimensional, i.e. valence and arousal. Hence, the adapted model looks like the following:

$$\text{Model \#1:} \quad \left. \begin{array}{l} V_t = w_{v,0} + w_{v,S} \sum_{j=1}^{t} \gamma_v^{t-j} * S_{v,j} \\[2em] A_t = w_{a,0} + w_{a,S} \sum_{j=1}^{t} \gamma_a^{t-j} * S_{a,j} \end{array} \right\} \tag{2.1}$$

where $V_t$ and $A_t$ are experienced valence and arousal at time point $t$, respectively (that is, after the $t$th image within an experimental block). $S_{v,j}$ and $S_{a,j}$ are the normative valence and arousal of the $j$th stimulus the individual encounters, respectively. The free parameters of the model in equation (2.1) are $w$ and $\gamma$ terms determining the integration of affective information. $\gamma_v$ and $\gamma_a$ are forgetting factors with respect to valence and arousal dimensions adjusting the influence of recent events in comparison with earlier events with $0 \leq \gamma \leq 1$. This parameter defines the relative impact of earlier versus later stimuli on momentary affect. As $\gamma$ approaches 1, each stimulus is weighted evenly; as it approaches 0, only the most recent stimulus determines currently experienced affect. Further, $w_{v,S}$ and $w_{a,S}$ are weights on the normative valence and arousal of the stimuli, representing the degree to which the stimuli impact affective experience. Finally, $w_{v,0}$ and $w_{a,0}$ are constant terms in valence and arousal dimensions, around which an individual's affective state fluctuates.

We selected the ED model as a candidate model because it has been previously used to investigate momentary subjective affective state (i.e. trial-to-trial happiness ratings) in a risky choice task. Nevertheless, the model has not been applied to an attentional task with visual images. Therefore, it is important to study how well the model generalizes to a setting other than the risky choice paradigm. We fit the model in equation (2.1) separately for each participant's valence and arousal ratings during the control block (19 affect ratings, 76 images). The model for the task blocks (76 affect ratings, 304 images) contained separate weight parameters and forgetting factors for task-relevant and task-irrelevant stimuli to study the potential differences in the model as a function of stimulus relevance.

### 2.3.3.2. Model #2 (weighted impact)

The other candidate model, *weighted impact* (WI), aimed to account for the momentary change in an individual's affective state between two consecutive time points prompted by an encountered stimulus. Hence, we formulated this model attempting to capture the continuous integration of the affective impact of the current sensory input with the affective state prior to encountering the current stimulus. The affective impact prompted by a stimulus is represented as the weighted difference between the input's normative tendency to induce affect and the individual's current affective experience. This makes the model mathematically similar to the Rescorla–Wagner learning rule [32].

$$\left. \begin{array}{l} V_t = V_{t-1} + \beta_v * (S_{v,t} - V_{t-1}) \\ A_t = A_{t-1} + \beta_a * (S_{a,t} - A_{t-1}) \end{array} \right\} \tag{2.2}$$

where $V_{t-1}$ and $V_t$ (similarly $A_{t-1}$ and $A_t$) are affective experience at two consecutive time points. $S_{v,t}$ and $S_{a,t}$ are the normative valence and arousal of the $t$th image, respectively. The second terms in equation (2.2) represent how much weight ($\beta_v$ and $\beta_a$) is assigned to prompt a change in momentary affective experience. If this model is feasible, $\beta$ parameters representing the affective impact of stimuli should have the following constraint, $0 \leq \beta \leq 1$. The smaller the $\beta$, the smaller the affective impact of the incoming stimulus; hence, a $\beta$ of zero implies that the stimuli have no impact on affective experience, while a $\beta$ of one means that the affective experience at a given time depends only on the most recent stimulus. In addition, similar to $\gamma$ parameter in the ED model, $\beta$ parameter determines how much of the impact of earlier stimuli are represented in affective experience. A high $\beta$ implies that even though the affective impact of current information is strong, the affective state is quickly updated with stimuli that follow. By contrast, a low $\beta$ means that the impact of the current stimulus is weak, but it will carry over relatively longer [33].

The WI model represents current affective experience as the integration of the affective impact of currently active information with previous affective experience. Hence, $\beta$ parameter captures the relative contribution of the previous affective state and the current sensory stimulation on currently experienced affect. For an easier comparison with the ED model, we can redefine the WI model to represent the cumulative affective impact of encountered stimuli in the following form:

$$\text{Model \#2:} \quad \left. \begin{array}{l} V_t = V_0 * (1 - \beta_v)^t + \sum_{j=1}^{t} \beta_v * (1 - \beta_v)^{t-j} * S_{v,t} \\[2ex] A_t = A_0 * (1 - \beta_a)^t + \sum_{j=1}^{t} \beta_a * (1 - \beta_a)^{t-j} * S_{a,t} \end{array} \right\} \tag{2.3}$$

where $V_t$ and $A_t$ reflect affective state (valence and arousal) at time point $t$, while $V_0$ and $A_0$ represent the initial affective state of the individual at the start of the block. During the study, the affective impact of

images are continuously integrated with affective experience. When compared with equation (2.1), the integration parts (i.e. the sum operations) of the two models are mathematically similar. The integration of the impact of stimuli in the WI model is represented by one parameter ($\beta$), in contrast to two in the ED model ($w_s$ and $\gamma$). However, the similarities between the models are limited to this as the first terms in equations (2.1) and (2.3) are different. Critically, the main difference between the models is how trial-to-trial affective experience is generated. According to the ED model, individuals, at each time point, integrate the affective impact of previously encountered events, and this process is determined by a forgetting factor discounting the impact of earlier stimuli and specific integration weights for the stimuli. Whereas the WI model assumes that, at a given time point, individuals integrate the affective impact of currently active information with their prior affective state.

We fit the WI separately for individual valence and arousal ratings during the control block. Then, we introduced separate $\beta$ parameters for task-relevant and task-irrelevant images for the task blocks to study differences due to stimulus relevance. The model fits predicted $\beta$ parameters and the initial values of valence and arousal ratings for each individual ($V_0$ and $A_0$ in equation (2.3)).

### 2.3.3.3. Model comparison

Model fits were carried out to maximize model log-likelihoods, which were used to compute model evidence based on Akaike information criterion (AIC), estimated for each participant and model. These AIC values were then submitted to a group-level random-effect analysis [34] that evaluates protected exceedance probability (*pxp*) representing the likelihood that a given model is more frequently used by participants than the other compared models. The *pxp* calculation was carried out using *spm_BMS* function implemented in SPM12.

## 3. Results

### 3.1. Task metrics

The average individual response rates for relevant and irrelevant target-images were 95.6% and 13.8%, respectively. The average individual hit rates (correctly identified targets in response trials) were 95.7% and 92.3% for relevant and irrelevant target-images. There were no false alarms during no-target trials. This pattern of results suggest that participants understood the task instructions and performed accordingly. They responded to a much smaller extent during task-irrelevant stimuli. However, when they chose to respond hit rates were comparable.

RTs (correct trials only) were analysed with a GLMM containing fixed effects of normative valence and arousal of the image and block order. We found that slower correct responses were associated with less pleasant ($B = -4.97$, 95CI = [$-8.85$, $-1.09$], $p = 0.012$) and more arousing images ($B = 4.22$, 95CI = [0.32, 8.12], $p = 0.034$). Moreover, RTs were longer for later blocks ($B = 3.99$, 95CI = [0.52, 7.45], $p = 0.024$).

### 3.2. Behavioural results

GLMMs predicting experienced valence and arousal contained fixed effects of normative valence and arousal of the four images presented prior to each affective rating. The images made significant contributions to self-reported valence with positive and significant coefficient estimates (figure 2). The relative contribution of an image was higher when it was presented later in a sequence, which points to a weighted-averaging mechanism that assigns higher weights to more recently encountered stimuli. Images also made significant contributions to self-reported arousal with positive coefficient estimates (figure 2). The coefficient estimates in the arousal model were lower in comparison with those in the valence model. Taken together, these results indicate that affective impact of the given stimuli are temporally integrated to influence an individual's momentary affective experience.

Importantly, we studied the impact of image relevance on momentary affect during the task blocks using GLMMs containing fixed effects of normative image affect and the interaction between the image affect and relevance. The normative valence of all the four stimuli were positively associated with experienced valence with higher coefficient estimates for more recent images (table 1). The last two images made significant contributions to experienced arousal. Interestingly, the interaction between image affect and relevance was significant for the last two images on valence dimension ($Image_{t-1}$: $B = 0.049$, 95CI = [0.004, 0.094], $p = 0.032$; $Image_{t-2}$: $B = 0.069$, 95CI = [0.024, 0.114], $p = 0.003$).

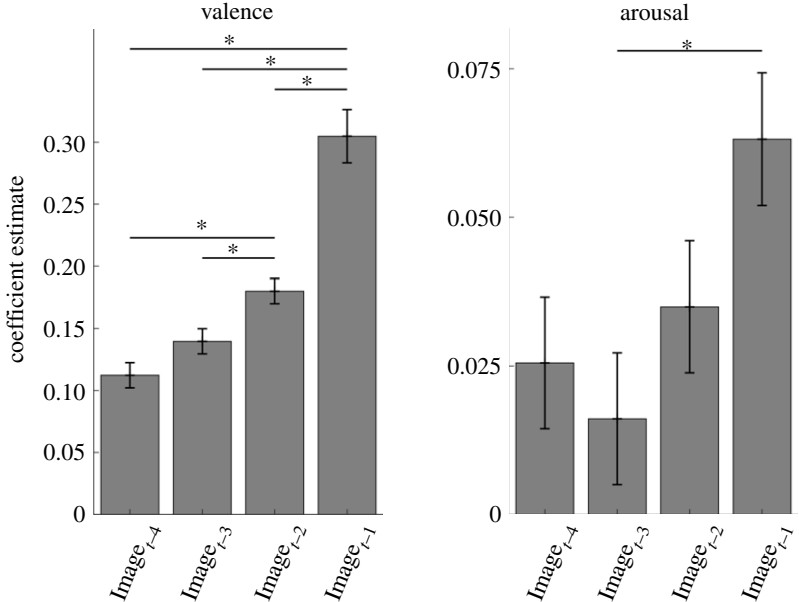

**Figure 2.** Results of GLMM analysis predicting valence and arousal ratings as a linear combination of normative affect of the images viewed prior to an affect rating. Coefficient estimates and standard errors are presented. Wald tests were used to compare the coefficient estimates. Holm–Bonferroni corrections were applied. (Image$_{t-1}$ to Image$_{t-4}$: the four images prior to an affect rating.) $^*p < 0.05$.

**Table 1.** The influence of normative image affect and relevance on self-reported valence and arousal. Note: coefficient estimates and 95% confidence intervals (numbers in brackets) are presented. Image$_{t-1}$ to Image$_{t-4}$: four images prior to an affect rating.

| model parameters | valence model | arousal model |
| --- | --- | --- |
| (intercept) | −0.001 [−0.023, 0.021] | −0.000 [−0.024, 0.024] |
| Image$_{t-4}$ | 0.074 [0.042, 0.106]** | 0.016 [−0.019, 0.051] |
| Image$_{t-3}$ | 0.113 [0.075, 0.151]** | 0.015 [−0.019, 0.049] |
| Image$_{t-2}$ | 0.131 [0.092, 0.169]** | 0.048 [0.014, 0.083]* |
| Image$_{t-1}$ | 0.271 [0.222, 0.32]** | 0.051 [0.016, 0.085]* |
| Image$_{t-4}$ × relevance | 0.03 [−0.015, 0.075] | 0.017 [−0.032, 0.067] |
| Image$_{t-3}$ × relevance | 0.021 [−0.024, 0.066] | −0.018 [−0.067, 0.031] |
| Image$_{t-2}$ × relevance | 0.069 [0.024, 0.114]* | −0.036 [−0.085, 0.013] |
| Image$_{t-1}$ × relevance | 0.049 [0.004, 0.094]* | 0.027 [−0.022, 0.076] |

$^*p < 0.05$, $^{**}p < 0.001$.

This indicates that relevance boosted the affective impact of stimuli on experienced pleasantness. However, this effect was limited in its temporal extent as it did not extend to more than two stimuli.

## 3.3. Computational models

We examined trial-by-trial fluctuations in affective experience using the normative affective impact and task-relevance of stimuli with two computational models. We sought to determine which of the candidate models best fitted the data and investigate the effect of stimulus relevance on key dynamic parameters of momentary affective experience. We compared the relative fit of the candidate models in control and task blocks separately using Bayesian model selection [34].

### 3.3.1. Control block

The two candidate models were fitted to valence and arousal responses during the control block. We have carried out a model recovery, wherein 100 datasets (valence and arousal ratings, $N = 83$,

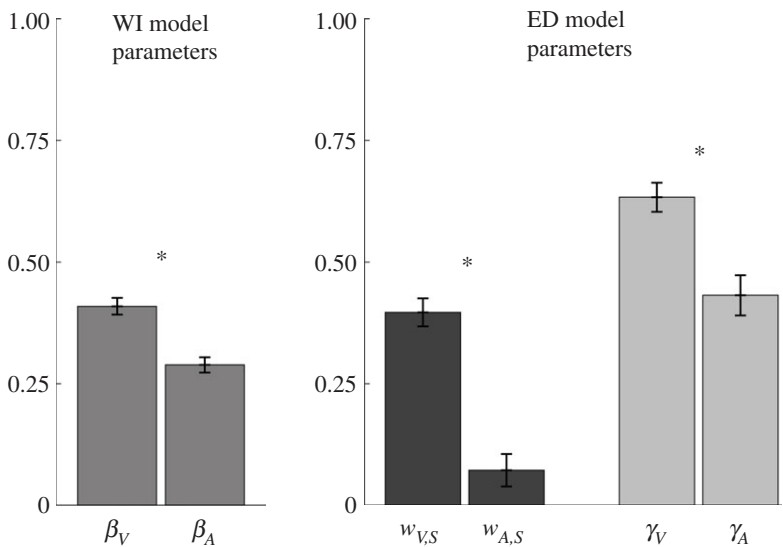

**Figure 3.** Comparison of the model parameters fitted to self-reported valence and arousal during the control block. $\beta_V$ and $\beta_A$ parameters (update parameter on valence and arousal) from the WI model are plotted on the left panel, whereas parameters of the ED model are plotted on the right ($w_{V,S}$: stimulus weight on valence; $w_{A,S}$: stimulus weight on arousal; $\gamma_V$: forgetting factor on valence; $\gamma_A$: forgetting factor on arousal). Error bars represent standard errors. $^*p < 0.05$.

no. of trials = 19) were simulated from each model using a range of randomly selected parameters (for details, see electronic supplementary material, figure S1 and Results). Both models were fitted to all 200 simulated datasets. The results show that the probability that each model is the best fit to data generated from the other model was very low (electronic supplementary material, figure S1): $p(fit = ED \mid simulated = WI) = 0.01$; $p(fit = WI \mid simulated = ED) < 0.01$. This demonstrates the ability of the current analysis to distinguish between different models under ideal conditions.

Two individuals were removed from arousal predictions since they responded without moving the slider on the arousal scale in all control block trials. According to Bayesian model selection, *weighted impact* (WI) was the best model (electronic supplementary material, table S1) to account for fluctuations in self-reported valence ($pxp = 0.998$) and arousal ($pxp > 0.999$). This finding suggests that WI is a plausible model for explaining momentary changes in affective experience (for the distribution of model parameters, see electronic supplementary material).

$\beta$ parameter in valence model was significantly higher in comparison with arousal; $t_{80} = 6.35$, $p < 0.001$ (figure 3). This finding indicates that sensory input prompted larger changes in experienced valence than in experienced arousal, a pattern that was also found in statistical analysis of the data (see figure 2 and table 1). Moreover, we examined the parameters of the ED model, even though it performed worse than the WI model. We found that both weight ($w$; $t_{80} = 7.4$, $p < 0.001$) and forgetting ($\gamma$; $t_{80} = 4.3$, $p < 0.001$) parameters were significantly higher in valence compared with arousal (figure 3). These results also confirm the pattern of coefficient estimates in the statistical analysis of the data: (i) the normative affective impact of the stimuli had a higher contribution to experienced pleasantness compared with experienced arousal; and (ii) the relative contribution of previously encountered images decreased with time and this decay was stronger in valence than in arousal ratings.

### 3.3.2. Task blocks and the effect of stimulus relevance

The ability of the model fitting to distinguish between the two models in the task blocks were assessed using simulations. One hundred datasets were simulated from each model (valence and arousal ratings, $N = 83$, no. of trials = 76). Then, both models were fitted to all 200 simulated datasets. The model that generated the data was the best fit in all 200 simulations (electronic supplementary material, figure S1), which demonstrates the ability for the analysis to distinguish between the different models.

We fitted the computational models to valence and arousal responses during the task blocks to investigate the effect of stimulus relevance on the fluctuations of momentary affective experience. According to Bayesian model selection, WI was the best model (electronic supplementary material, table S1) to account for fluctuations in self-reported valence and arousal ($pxp > 0.999$).

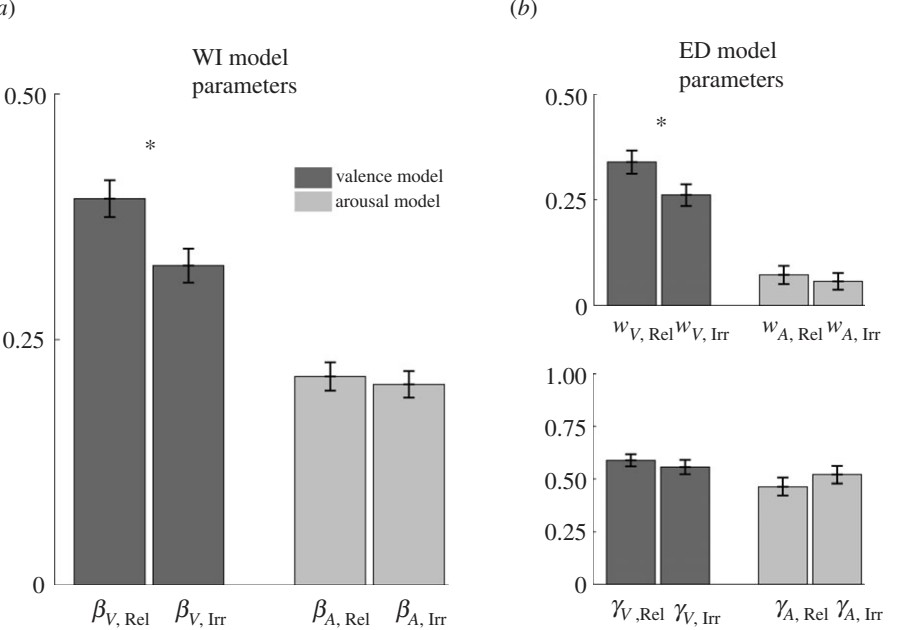

**Figure 4.** Comparison of the model parameters fitted to valence and arousal during the task blocks. (*a*) The parameters of the WI model ($\beta_{V, \text{Rel}}$: the impact of a task-relevant image on valence; $\beta_{V, \text{Irr}}$: the impact of a task-irrelevant image on valence; $\beta_{A, \text{Rel}}$: the impact of a task-relevant image on arousal; $\beta_{A, \text{Irr}}$: the impact of a task-irrelevant image on arousal). The comparisons showed that task-relevant images had a stronger impact than task-irrelevant images on valence. (*b*) The parameters of the ED model ($w_{V, \text{Rel}}$, $w_{A, \text{Rel}}$: the weight of task-relevant images on valence and arousal; $w_{V, \text{Irr}}$, $w_{A, \text{Irr}}$: the weight of task-irrelevant images on valence and arousal; $\gamma_{V, \text{Rel}}$, $\gamma_{A, \text{Rel}}$: the forgetting factors for task-relevant images on valence and arousal; $\gamma_{V, \text{Irr}}$, $\gamma_{A, \text{Irr}}$: the forgetting factors for task-irrelevant images on valence and arousal). Error bars represent standard errors. $^{*}p < 0.05$.

Next, we examined the differences in model parameters depending on stimulus relevance (figure 4). Critically, $\beta$ parameter for task-relevant images ($M = 0.32$, s.e. $= 0.02$) was significantly higher than for task-irrelevant images ($M = 0.25$, s.e. $= 0.02$) in valence predictions ($t_{82} = 4.09$, $p < 0.001$; figure 4*a*), which indicates that task-relevant images had a stronger impact on experienced pleasantness compared with task-irrelevant images. On the other hand, $\beta$ parameter difference due to stimulus relevance in arousal model did not reach statistical significance ($p > 0.25$). Furthermore, when we examined the parameters of the ED model, we have not found any significant differences in model parameters based on stimulus relevance in arousal dimension ($p > 0.25$). However, on valence dimension, the weight parameter was significantly higher for task-relevant compared with task-irrelevant stimuli ($t_{82} = 3.03$, $p = 0.003$; figure 4*b*), whereas the difference in gamma parameter due to stimulus relevance was not significant ($p > 0.25$). Taken together, the results from the computational models indicate that the images rendered relevant by the task instructions had a greater impact on experienced pleasantness in comparison with task-irrelevant images.

The computational modelling indicated that the WI model performed better than the ED model in explaining the variation in trial-to-trial affective experience. To verify that the important behavioural effects are captured by the model, we have validated the WI model using simulated data. One hundred datasets were generated from the WI model using the fit parameter values. We then analysed the simulated datasets using the same GLMMs we have used to assess the impact of normative image affect and image relevance on momentary affective experience (table 1). Figure 5 shows the distribution of coefficient estimates from the simulations together with the estimates and confidence intervals obtained from the observed data. The distributions depicted in figure 5 indicate that pattern of behavioural effects observed in the data are captured by the model since the range of estimates found in the simulated data overlaps with the observed coefficient estimates.

## 4. Discussion

The current research, conceptualizing the construction of affect as a temporally dependent and continuous process, set out to investigate the role of stimulus relevance on trial-by-trial fluctuations of

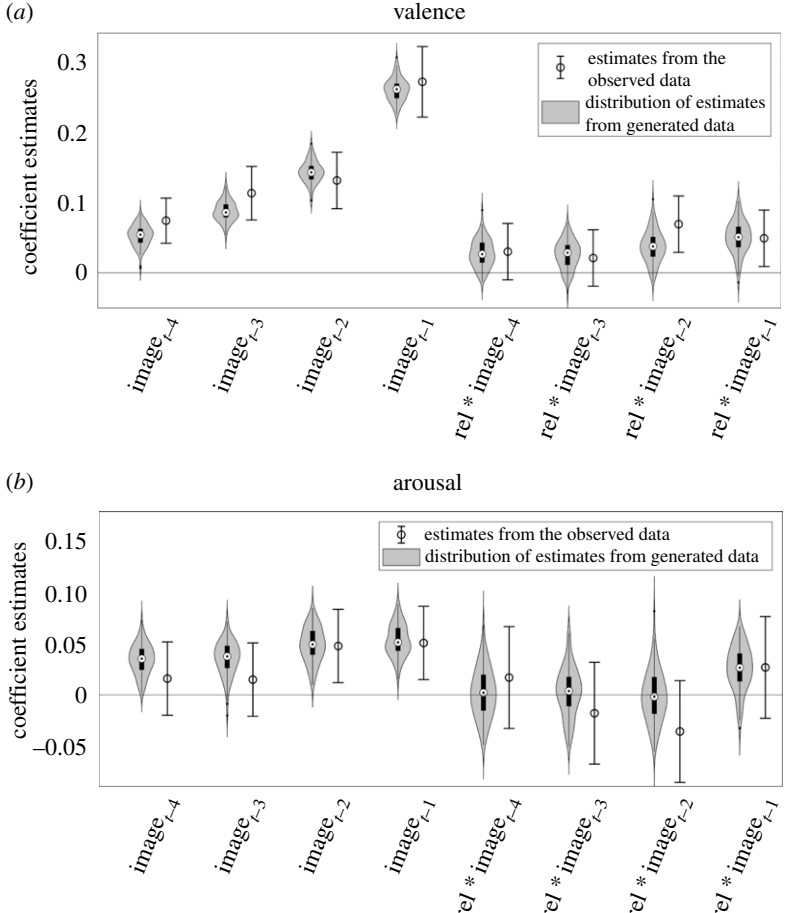

**Figure 5.** Validation of the WI model during the task blocks. One hundred datasets were generated with the fit values. All the stimulated datasets were analysed with the GLMMs used to analyse the original data. Violin plots show the distribution of the coefficient estimates (image affect and image relevance) from the simulated data. The error bars indicating the coefficient estimates and 95% confidence intervals observed in the data are also plotted for ease of comparison.

affective experience. The stimulus relevance was manipulated through a simple attentional task that rendered stimuli as task-relevant or task-irrelevant, controlling for the normative affective impact of the stimuli. Firstly, the current findings showed that affective experience at a given time is shaped by the temporal integration of the affective impact of currently active stimuli with previously experienced affect. Importantly, we show that stimulus relevance is a key factor represented in affective dynamics, that is, task-relevant stimuli, independent of stimulus affect, prompted larger fluctuations in experienced pleasantness compared with task-irrelevant stimuli in a dynamic context.

We have fitted and evaluated two computational models to explain the fluctuations in valence and arousal ratings. These models differed in their conceptualization of momentary affective state. The ED model generated the subjective affective state at a given time based on the weighted integration of previously encountered stimuli [28]. Whereas the WI model generated the subjective affective experience as a weighted integration of the previous affective experience and the currently active information, in which change from the previous affective experience is modelled as the weighted difference between the current stimulus' normative tendency to induce affect and the individual's previous affective experience. Both models attempt to capture the temporal integration of affective information with different parametrizations. The parametrization of the ED model is based on the affective integration of all previously encountered stimuli, while the WI model is based on a continuous integration, in which the affective impact of each stimulus is accounted for continuously. This means that all previously encountered stimuli are already represented in affective experience, to which the impact of currently active information is introduced. In other words, the ED model parametrizes the integration of the affective impact of the stimuli, while the WI model focuses on the continuous updating of affective experience. According to Bayesian model selection, the WI model

was the best model to explain the variation in self-reported affect during both control and task blocks. This finding indicates that the WI model presents a simple and plausible mechanism to capture the key dynamic parameters to explain trial-by-trial fluctuations in affective experience.

Critically, the current study clearly demonstrated that task-relevant images prompted larger changes in experienced pleasantness compared with task-irrelevant images. This finding suggests that relevance of events is a determining factor in moment-to-moment changes in affective experience. We navigate our environment with goal-directed action and behavioural relevance of objects is a critical feature of our sensory environment influencing sensory processing and attentional selection [14]. Previous research showed that a priority map representing behavioural importance and salience of objects inform attentional selection and control [11,15], and goal-relevance of stimuli modulates feature representations in attentional networks in the brain [20]. There is also an abundance of research on the relationship between affect and attention focusing on how affective salience of information (e.g. reward or threat) and experienced affect influence attentional selection (e.g. [21–24]). However, the current study is concerned with the reverse causal relationship and presents novel evidence that voluntary attentional selection history manipulated through stimulus relevance is represented in affective experience in a dynamic context. We studied stimulus relevance as a factor shaping affective experience, while controlling for affective salience of stimuli. The computational models of affective experience clearly indicate that goal-relevance of sensory stimuli is a determining factor of how a stream of sensory input is dynamically represented in momentary affect.

According to recent influential models, affect is linked to sensory (interoceptive) changes within the body's physiological systems [1,35]. These changes occur because the brain runs a predictive internal model of the world and represents physiological adaptations in response to changing environmental circumstances that are relevant to the individual. Therefore, the relevance of sensory stimuli defined by task instructions influencing attentional control and internal representation of affective consequences of the sensory environment is a biologically plausible mechanism for interpreting the current findings. Arguably, the current relevance manipulation modulated attentional sampling for task-relevant stimuli, which in turn led to increased impact of these events on momentary affect. These findings are also consistent with research showing that exogenously manipulated attention to an affective stimulus changes the affective experience of that same stimulus [26] as well as research showing that attentional deployment is one of the fundamental mechanisms behind affect regulation (e.g. [36]). Taken together, the current research suggests that momentary affective experience is constructed from the continuous integration of the affective impact of the currently active information with previously experienced affect, and that the behavioural relevance of stimuli is a key factor shaping this dynamic integration process.

The pattern of the fitted parameters in both models as well as the statistical analyses showed that stimuli typically prompted larger changes in experienced valence compared with arousal and that the recency effect was stronger in valence than in arousal. Taken together, these findings indicate that experienced arousal did not fluctuate as much as experienced valence did, a finding in line with previous studies [9,10]. This can be explained with the fact that valence is a fundamental feature of human experience, and humans can easily differentiate pleasant and unpleasant affect but differentiating high and low arousal is not ubiquitous. Hence, individuals' lower sensitivity to discriminate arousal states is one potential explanation for the current findings. Moreover, it is important to note that the set of affective images in the current experimental context might not have been successful in inducing a wide range of arousal experiences, which would automatically lead to relatively smaller $\beta$ parameters in WI and $w$ parameters in ED models (figure 3).

The relevance manipulation employed in the current study was aimed to study the influence of voluntary attentional selection on affective experience. An interesting possible extension of the findings is the impact of involuntary attentional selection, which was out of the current scope. An experimental design crossing task-relevance (voluntary attentional selection) and salience (involuntary attentional selection) conditions may be appropriate to study the affective impact of both voluntary and involuntary attentional selection. Finally, the current relevance manipulation (i.e. detecting visual targets on affective stimuli) may seem like a distractor rather than engagement with the affective dimensions of stimuli, which would reduce the correlations between the normative image ratings and affective experience for task-relevant images. However, the findings show that task-relevant images are weighted more heavily in comparison with task-irrelevant images. We propose that the trial structure may offer some explanations. The targets that were small compared with the images (less than 1% of the image size) were presented at a random location and at least 1500 ms after the image onset. This probably induced a visual search mode and modulated the attentional sampling of the

image. We believe that this specific trial structure ensured the differential attentional sampling of task-relevant and task-irrelevant images leading to an increased impact of task-relevant images on affective experience.

Ethics. Ethical approval for the current work is obtained from the Swedish Ethical Review Authority (registration no. 2019-03898).

Data accessibility. All data and analysis codes are available at https://osf.io/nhw3k/.

Authors' contributions. E.A. and D.V. developed the study concept and design. E.A. conducted data collection and analysis and drafted the manuscript. D.V. provided critical revisions. Both authors approved the final version of the manuscript for submission.

Competing interests. The authors have no competing interests.

Funding. This work is supported by the Swedish Research Council (grant no. 2017-02470). Funders had no role in study design, data collection, analysis, decision to publish or preparation of the manuscript.

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
