## [Peer Review File · Royal Society Open Science]

Review History

RSOS-210525.R0 (Original submission)

Review form: Reviewer 1 (Lei Zhang)

Is the manuscript scientifically sound in its present form?

No

Are the interpretations and conclusions justified by the results?

No

Is the language acceptable?

Yes

Do you have any ethical concerns with this paper?

No

Have you any concerns about statistical analyses in this paper?

Yes

Recommendation?

Major revision is needed (please make suggestions in comments)

Comments to the Author(s)

Comments to the Author:

Asutay & Västfjäll followed up and extended their previous work (Asutay et al., 2020, 2021) to examine how task-relevant affective stimuli impact individuals' moment-by-moment affective experience, ie, valence and arousal in the present study. Using GLM, the authors found that the most recent affective stimuli had the strongest impact on the affective experience (on rating scales), and using computational modeling (2 candidate models), they reported that affective experience is temporally updated by integrating the difference between the affective properties of the observed stimuli and the previous affective experience. Overall, this work is interesting, and the paper is well-written. However, I do have some concerns of the experimental design and I am not entirely convinced by the modeling analysis. Hopefully, these comments below would help to improve the paper.

Major points.

1. I wonder why after *every 4th image* participants were asked to provide their affective experience? There is no randomization, hence participants would formulate some expectation exactly every 4th image. If the participants would foresee/predict the upcoming rating, could it be possible that their reported behavior is somewhat regulated, rather than genuine affective experience? Plus, are task-relevant vs. task-irrelevant images intermixed within each mini block (ie those 4 images in-between participants ratings)? And how are the task-relevant vs. task-irrelevant images randomized/counterbalanced?
2. I am not sure if the control study is a good control. The control here differs from the main task in two aspects: (1) no task relevant vs irrelevant, (2) no target. But ideally, only one aspect ought to differ. A better control would be, participants still have to identify the targets, but now for all the images.
3. In the first analysis, hits and false alarm were analyzed. But according to signal detection theory, miss and correct rejection should also be included. Or perhaps, a d-prime score is a better index.
4. In the GLM analysis, it would be great to see the results separating task-relevant vs. task-irrelevant images. This way, it is more consistent with the modeling analysis.
5. For the exponential decay (ED) model, I think it makes better sense to decay over only the previous 4 images before each behavioral rating, rather than decaying over the entire trials (=76! for the last rating). If the authors' theoretical reasoning is to test a more discrete integration in the ED model, then a refresh/reset after each affective experience rating could better capture this.
6. Multiple comments on the Weighted Impact (WI) model.
 - 6a. The model specification seems to reflect the Rescorla-Wagner learning rule. Please consider citing the original work (Rescorla & Wagner, 1972).
 - 6b. The interpretation of the weight parameter (beta) is not entirely precise. The authors explained the meaning of beta == 0 and beta == 1, but in fact, any beta in-between 0 and 1 captures a balance between the weight on the most recent stimulus and the carry-over effect of previous stimuli. Please consider the following primer on this topic (Zhang et al., 2020).
 - 6c. My main concern of this model is which affective stimuli are considered? Looking at Equation (2), it seems that only the image immediately precedes participants' rating is used in the model; but how about the previous 3 images? If the ED model uses all the images, shouldn't it be the

same that WI also uses all the images? A simple fix is to take the average of the valence/arousal of the 4 images, to replace the current $S_{v,t}$ and $S_{a,t}$.

7. The model comparison result is confusing (Table S1). If the standard AIC calculation was used, then shouldn't it be that lower AIC indicates better model? But here the AIC results contradict the ppx results. Please provide more details of the AIC and ppx calculation, and if it is indeed the case, it would be helpful if the authors could discuss this contradicted data.

8. When interpreting the results that task-relevant images had stronger impact on behavior, I wonder how does task-relevant differ from attention? Perhaps a follow-up study that additionally includes a task-irrelevant but high attention involvement condition might help. It will be helpful to discuss this as a possible future direction.

Minor points.

- P2, there is an additional "a" in the title.
- P3, the second paragraph seems rather repetitive and redundant. Please shorten or simply remove it.
- P6-L39, the description of the power analysis is inaccurate. Power = 0.95 does not necessarily mean $\alpha = 0.05$. Power is $1 - \beta$, yet α is the significance level.
- P6-L42, I wonder why the study recruited 89 participants when the power analysis suggested 54? What stopping rule was used?
- P7-L20, "Participants' task was to identify the target [...]", it is better to explicitly say to identify the *shape* of the target. I was initially confused by what it means to "identify" the target.
- In a few places, the term "rating" actually means the OASIS rating (eg, P9-L48; P10-L44), rather than the behavioral rating by the participants. This created some confusion, so please change it throughout.
- It might make better sense to swap Fig 1A and 1B.
- It would be helpful if the authors could create a readme file (instructing how to use the code etc.) on OSF, to facilitate easy replication of the results.

Reference:

- Rescorla, R.A., and Wagner, A.R. (1972). A theory of Pavlovian conditioning: Variations in the effectiveness of reinforcement and nonreinforcement. In *Classical Conditioning II: Current Research and Theory*, (New York, NY: Appleton-Century-Crofts), pp. 64-99.
- Zhang, L., Lengersdorff, L., Mikus, N., Gläscher, J., & Lamm, C. (2020). Using reinforcement learning models in social neuroscience: frameworks, pitfalls and suggestions of best practices. *Social cognitive and affective neuroscience*, 15(6), 695-707.

Review form: Reviewer 2

Is the manuscript scientifically sound in its present form?

No

Are the interpretations and conclusions justified by the results?

No

Is the language acceptable?

Yes

Do you have any ethical concerns with this paper?

No

Have you any concerns about statistical analyses in this paper?

Yes

Recommendation?

Major revision is needed (please make suggestions in comments)

Comments to the Author(s)

The authors present a potentially interesting study on how different task-contexts can change affective experience. The research question as well as the study and task design are excellent, sufficiently novel, and of high relevance. Most of the methods are described properly but some important details and justifications are missing before I can properly evaluate this submission.

Major issues

1. The sample size appears a bit small for such a complex behavioral study even though it was justified with a power analysis. Of course, I appreciate that the authors cannot change this fact.
2. Models 1 and 2 are not properly motivated. It is not clear why exactly these two models were used, what their similarities are and where they differ. Under which conditions are they equivalent?
3. How exactly was the model selection performed? How was it implemented?
4. Why did the authors use the AIC to estimate the model evidence?
5. The authors should report some validation that their models reasonably explain the observed data, e.g., as in Rutledge et al. 2014, Fig 1B.

Critical issue

Several of these major issue add up to one critical issue that needs to be resolved before the usefulness of this work can be fully determined. If the power was not sufficient due to noisy data (issue 1) AIC would likely favor the less complex WI model. That's why other model criteria (issue 4) and reporting how the model fits the data (issue 5) would be of interest.

Minor issues

- 'The models also contained random intercepts and slopes at the participant level.' - Please explain.
- Figures 2-4 are unnecessarily hard to understand because of the uninformative labels and even less informative figure captions. Maybe Figure 1 and 2 could use the same labels and layout so it is clearer what img#1 to 4 refer to? WI and ED are introduced so they can be used in the figure (instead of model #2 and #1).
- There should be a better link between the GLMM and computational modeling results in the discussion.

Decision letter (RSOS-210525.R0)

Dear Dr Asutay

The Editors assigned to your paper RSOS-210525 "The goal-relevance of affective stimuli is dynamically represented in affective experience" have made a decision based on their reading of the paper and any comments received from reviewers.

Regrettably, in view of the reports received, the manuscript has been rejected in its current form. However, a new manuscript may be submitted which takes into consideration these comments.

We invite you to respond to the comments supplied below and prepare a resubmission of your manuscript. Below the referees' and Editors' comments (where applicable) we provide additional requirements. We provide guidance below to help you prepare your revision.

Please note that resubmitting your manuscript does not guarantee eventual acceptance, and we do not generally allow multiple rounds of revision and resubmission, so we urge you to make every effort to fully address all of the comments at this stage. If deemed necessary by the Editors, your manuscript will be sent back to one or more of the original reviewers for assessment. If the original reviewers are not available, we may invite new reviewers.

Please resubmit your revised manuscript and required files (see below) no later than 21-Feb-2022. Note: the ScholarOne system will 'lock' if resubmission is attempted on or after this deadline. If you do not think you will be able to meet this deadline, please contact the editorial office immediately.

Please note article processing charges apply to papers accepted for publication in Royal Society Open Science (<https://royalsocietypublishing.org/rsos/charges>). Charges will also apply to papers transferred to the journal from other Royal Society Publishing journals, as well as papers submitted as part of our collaboration with the Royal Society of Chemistry (<https://royalsocietypublishing.org/rsos/chemistry>). Fee waivers are available but must be requested when you submit your manuscript (<https://royalsocietypublishing.org/rsos/waivers>).

Thank you for submitting your manuscript to Royal Society Open Science and we look forward to receiving your resubmission. If you have any questions at all, please do not hesitate to get in touch.

on behalf of Dr Giorgia Silani (Associate Editor) and Essi Viding (Subject Editor)
openscience@royalsociety.org

Associate Editor Comments to Author (Dr Giorgia Silani):

Comments to the Author:

Two reviewers have found your work potentially interesting and innovative, but they have also raised several points, that need to be addressed before further consideration. Given the revision needs considerable work, the reject and allow resubmission option seems the most appropriate at this stage.

Reviewer comments to Author:

Reviewer: 1

Comments to the Author(s)

Comments to the Author:

Asutay & Västfjäll followed up and extended their previous work (Asutay et al., 2020, 2021) to examine how task-relevant affective stimuli impact individuals' moment-by-moment affective

experience, ie, valence and arousal in the present study. Using GLM, the authors found that the most recent affective stimuli had the strongest impact on the affective experience (on rating scales), and using computational modeling (2 candidate models), they reported that affective experience is temporally updated by integrating the difference between the affective properties of the observed stimuli and the previous affective experience. Overall, this work is interesting, and the paper is well-written. However, I do have some concerns of the experimental design and I am not entirely convinced by the modeling analysis. Hopefully, these comments below would help to improve the paper.

Major points.

1. I wonder why after *every 4th image* participants were asked to provide their affective experience? There is no randomization, hence participants would formulate some expectation exactly every 4th image. If the participants would foresee/predict the upcoming rating, could it be possible that their reported behavior is somewhat regulated, rather than genuine affective experience? Plus, are task-relevant vs. task-irrelevant images intermixed within each mini block (ie those 4 images in-between participants ratings)? And how are the task-relevant vs. task-irrelevant images randomized/counterbalanced?
2. I am not sure if the control study is a good control. The control here differs from the main task in two aspects: (1) no task relevant vs irrelevant, (2) no target. But ideally, only one aspect ought to differ. A better control would be, participants still have to identify the targets, but now for all the images.
3. In the first analysis, hits and false alarm were analyzed. But according to signal detection theory, miss and correct rejection should also be included. Or perhaps, a d-prime score is a better index.
4. In the GLM analysis, it would be great to see the results separating task-relevant vs. task-irrelevant images. This way, it is more consistent with the modeling analysis.
5. For the exponential decay (ED) model, I think it makes better sense to decay over only the previous 4 images before each behavioral rating, rather than decaying over the entire trials (=76! for the last rating). If the authors' theoretical reasoning is to test a more discrete integration in the ED model, then a refresh/reset after each affective experience rating could better capture this.
6. Multiple comments on the Weighted Impact (WI) model.
 - 6a. The model specification seems to reflect the Rescorla-Wagner learning rule. Please consider citing the original work (Rescorla & Wagner, 1972).
 - 6b. The interpretation of the weight parameter (beta) is not entirely precise. The authors explained the meaning of $\beta = 0$ and $\beta = 1$, but in fact, any β in-between 0 and 1 captures a balance between the weight on the most recent stimulus and the carry-over effect of previous stimuli. Please consider the following primer on this topic (Zhang et al., 2020).
 - 6c. My main concern of this model is which affective stimuli are considered? Looking at Equation (2), it seems that only the image immediately precedes participants' rating is used in the model; but how about the previous 3 images? If the ED model uses all the images, shouldn't it be the same that WI also uses all the images? A simple fix is to take the average of the valence/arousal of the 4 images, to replace the current $S_{v,t}$ and $S_{a,t}$.
7. The model comparison result is confusing (Table S1). If the standard AIC calculation was used, then shouldn't it be that lower AIC indicates better model? But here the AIC results contradict the ppx results. Please provide more details of the AIC and ppx calculation, and if it is indeed the case, it would be helpful if the authors could discuss this contradicted data.

8. When interpreting the results that task-relevant images had stronger impact on behavior, I wonder how does task-relevant differ from attention? Perhaps a follow-up study that additionally includes a task-irrelevant but high attention involvement condition might help. It will be helpful to discuss this as a possible future direction.

Minor points.

- P2, there is an additional “a” in the title.
- P3, the second paragraph seems rather repetitive and redundant. Please shorten or simply remove it.
- P6-L39, the description of the power analysis is inaccurate. Power = 0.95 does not necessarily mean $\alpha = 0.05$. Power is $1 - \beta$, yet α is the significance level.
- P6-L42, I wonder why the study recruited 89 participants when the power analysis suggested 54? What stopping rule was used?
- P7-L20, “Participants’ task was to identify the target [...]”, it is better to explicitly say to identify the *shape* of the target. I was initially confused by what it means to “identify” the target.
- In a few places, the term “rating” actually means the OASIS rating (eg, P9-L48; P10-L44), rather than the behavioral rating by the participants. This created some confusion, so please change it throughout.
- It might make better sense to swap Fig 1A and 1B.
- It would be helpful if the authors could create a readme file (instructing how to use the code etc.) on OSF, to facilitate easy replication of the results.

Reference:

Rescorla, R.A., and Wagner, A.R. (1972). A theory of Pavlovian conditioning: Variations in the effectiveness of reinforcement and nonreinforcement. In *Classical Conditioning II: Current Research and Theory*, (New York, NY: Appleton-Century-Crofts), pp. 64-99.

Zhang, L., Lengersdorff, L., Mikus, N., Gläscher, J., & Lamm, C. (2020). Using reinforcement learning models in social neuroscience: frameworks, pitfalls and suggestions of best practices. *Social cognitive and affective neuroscience*, 15(6), 695-707.

Reviewer: 2

Comments to the Author(s)

The authors present a potentially interesting study on how different task-contexts can change affective experience. The research question as well as the study and task design are excellent, sufficiently novel, and of high relevance. Most of the methods are described properly but some important details and justifications are missing before I can properly evaluate this submission.

Major issues

1. The sample size appears a bit small for such a complex behavioral study even though it was justified with a power analysis. Of course, I appreciate that the authors cannot change this fact.
2. Models 1 and 2 are not properly motivated. It is not clear why exactly these two models were used, what their similarities are and where they differ. Under which conditions are they equivalent?
3. How exactly was the model selection performed? How was it implemented?
4. Why did the authors use the AIC to estimate the model evidence?
5. The authors should report some validation that their models reasonably explain the observed data, e.g., as in Rutledge et al. 2014, Fig 1B.

Critical issue

Several of these major issue add up to one critical issue that needs to be resolved before the usefulness of this work can be fully determined. If the power was not sufficient due to noisy data

(issue 1) AIC would likely favor the less complex WI model. That's why other model criteria (issue 4) and reporting how the model fits the data (issue 5) would be of interest.

Minor issues

- 'The models also contained random intercepts and slopes at the participant level.' - Please explain.
- Figures 2-4 are unnecessarily hard to understand because of the uninformative labels and even less informative figure captions. Maybe Figure 1 and 2 could use the same labels and layout so it is clearer what img#1 to 4 refer to? WI and ED are introduced so they can be used in the figure (instead of model #2 and #1).
- There should be a better link between the GLMM and computational modeling results in the discussion.

===PREPARING YOUR MANUSCRIPT===

===PREPARING YOUR REVISION IN SCHOLARONE===

Author's Response to Decision Letter for (RSOS-210525.R0)

See Appendix A.

RSOS-211548.R0

Review form: Reviewer 1

Is the manuscript scientifically sound in its present form?

Yes

Are the interpretations and conclusions justified by the results?

Yes

Is the language acceptable?

Yes

Do you have any ethical concerns with this paper?

No

Have you any concerns about statistical analyses in this paper?

No

Recommendation?

Accept with minor revision (please list in comments)

Comments to the Author(s)

Comments to the Author:

The authors have done a good job of addressing my previous questions and concerns. I do have some additional comments, though, in the hope of further improving the paper (also to make it more accessible to non-specialists).

1. Regarding my previous comment #3, I follow the reasoning of not including d -prime. But I anticipate a handful of readers will have the same question/concern, so it would be great to explicitly state the reason (the one in the response letter) in the main text.
2. I insist it'd be beneficial to include the Rescorla & Wagner (1972) reference. As the authors acknowledged, the math is the same, although the learning rate is commonly "known as" a learning parameter, it is nothing more than an updating/weighting parameter. It also helps the readers to relate to the Rescorla-Wagner model - so for knowledgeable readers, they may decide to skip some details if they see the correct citation. Last, I see no need to properly introduce learning models.
3. For both the ED and WI model, the authors may want to add the likelihood function (ie. Normal distribution, as indicated by the `normpdf` function in OSF) in the main text.
4. For the WI model, please include the initial value of V_t and A_t in the main text.

Minor:

- I wonder why the authors decided on the power to be 0.95? This is not a common power value in the literature.

- Maybe I missed it - I still did not see the readme file (instructing how to use the code etc.) on OSF.

- If the paper Zhang & Lengersdorff et al., (2020) has helped the authors to better explain the beta parameter (page 13) and to derive the current equation-3 and its explanation (page 14), please consider citing it.

Zhang, L., Lengersdorff, L., Mikus, N., Gläscher, J., & Lamm, C. (2020). Using reinforcement learning models in social neuroscience: frameworks, pitfalls and suggestions of best practices. *Social cognitive and affective neuroscience*, 15(6), 695-707.

Review form: Reviewer 2

Is the manuscript scientifically sound in its present form?

Yes

Are the interpretations and conclusions justified by the results?

Yes

Is the language acceptable?

Yes

Do you have any ethical concerns with this paper?

No

Have you any concerns about statistical analyses in this paper?

No

Recommendation?

Accept as is

Comments to the Author(s)

I would like to thank the authors for their substantial revisions and improvements.

Decision letter (RSOS-211548.R0)

Dear Dr Asutay

On behalf of the Editors, we are pleased to inform you that your Manuscript RSOS-211548 "The goal-relevance of affective stimuli is dynamically represented in affective experience" has been accepted for publication in Royal Society Open Science subject to minor revision in accordance with the referees' reports. Please find the referees' comments along with any feedback from the Editors below my signature.

Please submit your revised manuscript and required files (see below) no later than 7 days from today's (ie 26-Oct-2021) date. Note: the ScholarOne system will 'lock' if submission of the revision is attempted 7 or more days after the deadline. If you do not think you will be able to meet this deadline please contact the editorial office immediately.

on behalf of Dr Giorgia Silani (Associate Editor) and Essi Viding (Subject Editor)
openscience@royalsociety.org

Associate Editor Comments to Author (Dr Giorgia Silani):
Comments to the Author:

Dear authors, the paper has now been reviewed by the previous reviewers. Both recommend acceptance. One of the reviewer has still few points that need to be addressed.

Please provide a rebuttal and revise the manuscript accordingly. If sufficiently addressed I will proceed with acceptance without additional round of review.

Reviewer comments to Author:
Reviewer: 2
Comments to the Author(s)

I would like to thank the authors for their substantial revisions and improvements.

Reviewer: 1
Comments to the Author(s)
Comments to the Author:

The authors have done a good job of addressing my previous questions and concerns. I do have some additional comments, though, in the hope of further improving the paper (also to make it more accessible to non-specialists).

1. Regarding my previous comment #3, I follow the reasoning of not including d-prime. But I anticipate a handful of readers will have the same question/concern, so it would be great to explicitly state the reason (the one in the response letter) in the main text.

2. I insist it'd be beneficial to include the Rescorla & Wagner (1972) reference. As the authors acknowledged, the math is the same, although the learning rate is commonly "known as" a learning parameter, it is nothing more than an updating/weighting parameter. It also helps the readers to relate to the Rescorla-Wagner model – so for knowledgeable readers, they may decide to skip some details if they see the correct citation. Last, I see no need to properly introduce learning models.

3. For both the ED and WI model, the authors may want to add the likelihood function (ie. Normal distribution, as indicated by the `normpdf` function in OSF) in the main text.

4. For the WI model, please include the initial value of V_t and A_t in the main text.

Minor:

- I wonder why the authors decided on the power to be 0.95? This is not a common power value in the literature.

- Maybe I missed it – I still did not see the readme file (instructing how to use the code etc.) on OSF.

- If the paper Zhang & Lengersdorff et al., (2020) has helped the authors to better explain the beta parameter (page 13) and to derive the current equation-3 and its explanation (page 14), please consider citing it.

Zhang, L., Lengersdorff, L., Mikus, N., Gläscher, J., & Lamm, C. (2020). Using reinforcement learning models in social neuroscience: frameworks, pitfalls and suggestions of best practices. *Social cognitive and affective neuroscience*, 15(6), 695-707.

===PREPARING YOUR MANUSCRIPT===

one version should clearly identify all the changes that have been made (for instance, in coloured highlight, in bold text, or tracked changes);

===PREPARING YOUR REVISION IN SCHOLARONE===

-- Ensure that your data access statement meets the requirements at <https://royalsociety.org/journals/authors/author-guidelines/#data>. You should ensure that you cite the dataset in your reference list. If you have deposited data etc in the Dryad repository, please only include the 'For publication' link at this stage. You should remove the 'For review' link.

-- If you are requesting an article processing charge waiver, you must select the relevant waiver option (if requesting a discretionary waiver, the form should have been uploaded, see 'File upload' above).

-- If you have uploaded any electronic supplementary (ESM) files, please ensure you follow the guidance at <https://royalsociety.org/journals/authors/author-guidelines/#supplementary-material> to include a suitable title and informative caption. An example of appropriate titling and captioning may be found at https://figshare.com/articles/Table_S2_from_Is_there_a_trade-off_between_peak_performance_and_performance_breadth_across_temperatures_for_aerobic_scope_in_teleost_fishes_/3843624.

Author's Response to Decision Letter for (RSOS-211548.R0)

See Appendix B.

Decision letter (RSOS-211548.R1)

Dear Dr Asutay,

It is a pleasure to accept your manuscript entitled "The goal-relevance of affective stimuli is dynamically represented in affective experience" in its current form for publication in Royal Society Open Science. The comments of the reviewer(s) who reviewed your manuscript are included at the foot of this letter.

Please remember to make any data sets or code libraries 'live' prior to publication, and update any links as needed when you receive a proof to check - for instance, from a private 'for review'

URL to a publicly accessible 'for publication' URL. It is good practice to also add data sets, code and other digital materials to your reference list.

on behalf of Dr Giorgia Silani (Associate Editor) and Essi Viding (Subject Editor)
openscience@royalsociety.org

Associate Editor Comments to Author (Dr Giorgia Silani):
Associate Editor
Comments to the Author:

The authors have addressed all the remaining comments, in a satisfactory way. The paper can be accepted in the current form.

Appendix A

Dear editors,

First, we would like to thank you for the opportunity to submit a revised draft of our research article entitled “The goal-relevance of affective stimuli is dynamically represented in affective experience.” Also, we thank reviewers for their insightful comments and suggestions to improve our article. Following your decision and the reviewers’ comments, we revised the manuscript taking into account all the major and minor points that were raised in the reviews.

In the following, we list how we addressed each specific comment from the reviewers.

Reviewer: 1

1. I wonder why after *every 4th image* participants were asked to provide their affective experience? There is no randomization, hence participants would formulate some expectation exactly every 4th image. If the participants would foresee/predict the upcoming rating, could it be possible that their reported behavior is somewhat regulated, rather than genuine affective experience? Plus, are task-relevant vs. task-irrelevant images intermixed within each mini block (ie those 4 images in-between participants ratings)? And how are the task-relevant vs. task-irrelevant images randomized/counterbalanced?

Author response: We understand that having participants report their affective experience after every 4th image may present some issues with respect to predictability. However, we argue that this does not pose a problem. First, in our earlier studies we have tested different stimulus group sizes (affective experience reported after every 4th or 6th stimulus) and stimulus durations (2, 3, or 4 seconds per image) and obtained similar results (Asutay et al., 2021; 2020). We found that independent from the stimulus set or duration, reported experience is partly determined by a temporal integration of the affective impact of images and prior affective experience reported at the end of the previous trial. Additionally, the fact that reported experience does not depend solely on the most recent stimulus, but it is instead determined by a combination of recently encountered stimuli indicates that affective consequences of the stimuli are integrated in overall affective experience. Furthermore, we have included detailed information on the randomization of task-relevant and task-irrelevant images in the methods section (see Materials, Experimental Design, and Procedure section). Task-relevant and task-irrelevant images were equally distributed between blocks, but they

were presented in a random order within each block. Therefore, relevant and irrelevant images were intermixed within each 4 images in between two affect ratings.

2. I am not sure if the control study is a good control. The control here differs from the main task in two aspects: (1) no task relevant vs irrelevant, (2) no target. But ideally, only one aspect ought to differ. A better control would be, participants still have to identify the targets, but now for all the images.

Author response: We agree that the control task may not be the optimal control condition for the main task. We included this block of trials in their current form because this is the from closest to previous studies. Because one of our main motivations for the control block was to replicate the earlier findings. Also, this block enabled us to study the computational modeling when the context did not include an attentional task. We included this explanation in the revised version of the manuscript (see Materials, Experimental Design, and Procedure section).

3. In the first analysis, hits and false alarm were analyzed. But according to signal detection theory, miss and correct rejection should also be included. Or perhaps, a d-prime score is a better index.

Author response: Miss and correct rejection were not included since they are the opposite of hit and false alarm rates, respectively. We agree that d-prime is generally a better index than hit rates. However, the task was easy, which led to high hit rates and no false alarms. This would only produce d-prime scores at the very high end of the spectrum. Additionally, since false alarm rates did not differ between conditions or individuals, d-prime becomes a direct function of hit rates. Therefore, we chose to keep the analysis simple and presented hit rates and false alarms.

4. In the GLM analysis, it would be great to see the results separating task-relevant vs. task-irrelevant images. This way, it is more consistent with the modeling analysis.

Author response: We agree that this was critically missing in the previous version of the manuscript. We thank the reviewer for pointing this out. To address this, we included an analysis investigating the effect of normative stimulus affect and stimulus relevance in the

same model (see Table 1). In addition, we used this analysis to validate the WI model (see Figure 5).

5. For the exponential decay (ED) model, I think it makes better sense to decay over only the previous 4 images before each behavioral rating, rather than decaying over the entire trials (=76! for the last rating). If the authors' theoretical reasoning is to test a more discrete integration in the ED model, then a refresh/reset after each affective experience rating could better capture this.

Author response: We have tested the model suggested by the reviewer as an alternative, in which only the last 4 images were taken into consideration. This restricted model did not fit the data better than the full model. We presented this analysis in the Supplementary Material.

6. Multiple comments on the Weighted Impact (WI) model.

6a. The model specification seems to reflect the Rescorla-Wagner learning rule. Please consider citing the original work (Rescorla & Wagner, 1972).

Author response: We agree that the WI model mathematically similar to reinforcement learning (RL), even though it was not directly inspired by it. The learning parameter in RL determines learning of the value of an external reward. In the current model, the weight parameter is not a learning parameter but an updating parameter, which defines how fast affective state fluctuates. However, we have not included a RL reference, as this would require introducing learning models, which would unnecessarily complicate the manuscript.

6b. The interpretation of the weight parameter (beta) is not entirely precise. The authors explained the meaning of $\beta = 0$ and $\beta = 1$, but in fact, any beta in-between 0 and 1 captures a balance between the weight on the most recent stimulus and the carry-over effect of previous stimuli. Please consider the following primer on this topic (Zhang et al., 2020).

Author response: To address this comment, we have extended the section in methods, in which we discussed the model and the interpretation of the weight parameter in detail.

6c. My main concern of this model is which affective stimuli are considered? Looking at Equation (2), it seems that only the image immediately precedes participants' rating is used in the model; but how about the previous 3 images? If the ED model uses all the images, shouldn't it be the same that WI also uses all the images? A simple fix is to take the average of the valence/arousal of the 4 images, to replace the current $S_{v,t}$ and $S_{a,t}$.

Author response: The model included all the images like the ED model. It took the initial affective experience (at the beginning of a block) and updated it according to normative stimulus affect after each stimulus. We agree that the equation 2 might have been misleading. To clarify, we presented the model in a form representing the cumulative affective impact of encountered stimuli (see Equation 3 in the revised manuscript).

7. The model comparison result is confusing (Table S1). If the standard AIC calculation was used, then shouldn't it be that lower AIC indicates better model? But here the AIC results contradict the p_{xp} results. Please provide more details of the AIC and p_{xp} calculation, and if it is indeed the case, it would be helpful if the authors could discuss this contradicted data.

Author response: The AICs presented in Table S1 were negative. The values mistakenly reflected: $(\log\text{-likelihood}) - (\text{number of model parameters})$. We corrected this, so now the values in Table S1 are: $-2 * (\log\text{-likelihood}) + 2 * (\text{number of model parameters})$. We thank the reviewer for catching this error.

8. When interpreting the results that task-relevant images had stronger impact on behavior, I wonder how does task-relevant differ from attention? Perhaps a follow-up study that additionally includes a task-irrelevant but high attention involvement condition might help. It will be helpful to discuss this as a possible future direction.

Author response: We employed the relevance manipulation to study the influence of voluntary attentional selection on affective experience. We argue that task-relevance does not significantly differ from voluntary attention, which we discuss in detail in the manuscript. However, we have not studied other attention effects such as involuntary attention. We have included a short section on this as a possible extension of the current findings.

Minor points.

- P2, there is an additional "a" in the title.

Author response: The typo is fixed.

- P3, the second paragraph seems rather repetitive and redundant. Please shorten or simply remove it.

Author response: We have integrated the first two paragraphs.

- P6-L39, the description of the power analysis is inaccurate. Power = 0.95 does not necessarily mean $\alpha = 0.05$. Power is $1 - \beta$, yet α is the significance level.

Author response: In the initial version of the article, we did not mean that α and power is connected. It was meant that power was set to .95 and the significance level to .05 independently. We revised the sentence to clarify the misunderstanding.

- P6-L42, I wonder why the study recruited 89 participants when the power analysis suggested 54? What stopping rule was used?

Author response: Prior to the experiment, we decided that data collection would be open for at least two weeks, during which no analysis was conducted. Then, we stopped the data collection since our sample size was well over the sample size suggested by the power analysis. We revised the manuscript to include this (see Participants section).

- P7-L20, "Participants' task was to identify the target [...]", it is better to explicitly say to identify the *shape* of the target. I was initially confused by what it means to "identify" the target.

Author response: The task description was revised according to the comment.

- In a few places, the term "rating" actually means the OASIS rating (eg, P9-L48; P10-L44), rather than the behavioral rating by the participants. This created some confusion, so please change it throughout.

Author response: We revised the manuscript to address the confusion.

- It might make better sense to swap Fig 1A and 1B.

Author response: Upon viewing the manuscript with fresh eyes, we agree. We have updated Figure 1 and swapped fig 1A and 1B.

- It would be helpful if the authors could create a readme file (instructing how to use the code etc.) on OSF, to facilitate easy replication of the results.

Author response: A readme file is uploaded on the OSF page.

Reviewer: 2

Major issues

1. The sample size appears a bit small for such a complex behavioral study even though it was justified with a power analysis. Of course, I appreciate that the authors cannot change this fact.

Author response: We followed a formal power analysis and a priori determined data collection stopping rule. We understand the sample size may appear small. However, the general behavioral results we obtained are consistent with previous research, which is also evident in the computational modeling.

2. Models 1 and 2 are not properly motivated. It is not clear why exactly these two models were used, what their similarities are and where they differ. Under which conditions are they equivalent?

Author response: In the revised manuscript, we have extended the presentation of the models. The ED model was selected because it has been used and validated to study momentary affective experience (trial-by-trial happiness ratings) during a risky choice task. The ED model assumes the integration of the affective impact of previously encountered stimuli at each time point. We chose the WI model to compare the ED model to a mechanism, in which the integration occurs continuously; that is, at each time point the affective consequences of the currently active information is integrated with the previous affective state of the individual. We presented theoretical motivation for using these models, as well as their similarities and differences in the revised version of the manuscript.

3. How exactly was the model selection performed? How was it implemented?

Author response: The models were fit separately for each participant, and the AIC was calculated. The individual model evidence based on AIC was submitted to the Bayesian model selection to calculate the *pxp*. We used *spm_BMS* function implemented in SPM12 for *pxp* calculation. We explained this in the revised manuscript (Model Comparison section) and in the supplement (Model Fit and Comparison section).

4. Why did the authors use the AIC to estimate the model evidence?

Author response: AIC is one of the standard information criteria used in the context of maximum likelihood estimation. We also considered BIC but since it penalizes model complexity even more than AIC, we did not use it.

5. The authors should report some validation that their models reasonably explain the observed data, e.g., as in Rutledge et al. 2014, Fig 1B.

Author response: In the revised version, we included a model validation for the better performing WI model (see the last paragraph of the results section and Figure 5).

Critical issue

Several of these major issue add up to one critical issue that needs to be resolved before the usefulness of this work can be fully determined. If the power was not sufficient due to noisy data (issue 1) AIC would likely favor the less complex WI model. That's why other model criteria (issue 4) and reporting how the model fits the data (issue 5) would be of interest.

Author response: To address this critical point, we revised the manuscript in several ways. First, we have included a key statistical analysis showing the observed behavioral effects in the data (Table 1 in the revised manuscript). Then, we have validated the better performing WI model using simulations. We simulated 100 datasets from the WI model using the fit parameter values. The simulated datasets were analyzed using the same GLMM. The results showed that the distribution of coefficient estimates from the simulations agree with the observed effects (Figure 5 in the revised manuscript). Additionally, we have included a

model recovery analysis to show that the model fitting and the experiment can distinguish between the two models under ideal conditions of simulated datasets (Figure S1 in the Supplementary Material). We hope that these revisions can address this critical comment.

Minor issues

- 'The models also contained random intercepts and slopes at the participant level.' - Please explain.

Author response: The GLMMs contained both fixed and random effects. We included the following sentence to explain. "Hence, both the intercept and the estimates of the predictors were allowed to vary across individuals."

- Figures 2-4 are unnecessarily hard to understand because of the uninformative labels and even less informative figure captions. Maybe Figure 1 and 2 could use the same labels and layout so it is clearer what img#1 to 4 refer to? WI and ED are introduced so they can be used in the figure (instead of model #2 and #1).

Author response: Following the reviewer's advice, we updated the figures.

- There should be a better link between the GLMM and computational modeling results in the discussion.

Author response: We included a model validation based on the GLMM results and discussed the link between the behavioral effects in the GLMM and the computational modeling results.

We appreciate your careful evaluation of the work and hope that this revision meets with your approval. Thanks to your and the reviewers' thoughtful comments the manuscript is, in our opinion, much improved.

Best regards,

Erkin Asutay (on behalf of coauthors)

Appendix B

Dear editors,

First, we would like to thank you for the acceptance of our research article entitled “The goal-relevance of affective stimuli is dynamically represented in affective experience.” Also, we thank reviewers for their insightful comments and suggestions to improve our article.

We have revised the manuscript according to the first reviewer’s suggestions.

Reviewer: 1

1. Regarding my previous comment #3, I follow the reasoning of not including d -prime. But I anticipate a handful of readers will have the same question/concern, so it would be great to explicitly state the reason (the one in the response letter) in the main text.

Author response: We have included the explanation from our previous response letter in the methods section (page 9 in the revised manuscript tracked changes version).

2. I insist it’d be beneficial to include the Rescorla & Wagner (1972) reference. As the authors acknowledged, the math is the same, although the learning rate is commonly “known as” a learning parameter, it is nothing more than an updating/weighting parameter. It also helps the readers to relate to the Rescorla-Wagner model – so for knowledgeable readers, they may decide to skip some details if they see the correct citation. Last, I see no need to properly introduce learning models.

Author response: We have included the reference and pointed out the mathematical similarity between the models (page 12 in the revised manuscript tracked changes version).

3. For both the ED and WI model, the authors may want to add the likelihood function (ie. Normal distribution, as indicated by the `normpdf` function in OSF) in the main text.

Author response: This information is also introduced in the methods section (page 10 in the revised manuscript tracked changes version).

4. For the WI model, please include the initial value of V_t and A_t in the main text.

Author response: During the model fitting we have predicted the initial valence and arousal as well. Thanks for pointing that this information was missing. We have included this in methods section (page 14 in the revised manuscript tracked changes version).

5. I wonder why the authors decided on the power to be 0.95? This is not a common power value in the literature.

Author response: We aimed to have sufficient power to be able to find a weak signal in the data.

6. Maybe I missed it – I still did not see the readme file (instructing how to use the code etc.) on OSF.

Author response: We have included a word file explaining how to use the code.

7. If the paper Zhang & Lengersdorff et al., (2020) has helped the authors to better explain the beta parameter (page 13) and to derive the current equation-3 and its explanation (page 14), please consider citing it.

Author response: We have included the reference (page 13 in the revised manuscript tracked changes version).

We appreciate your careful evaluation of the work. Thanks to your and the reviewers' thoughtful comments the manuscript is, in our opinion, much improved.

Best regards,

Erkin Asutay (on behalf of coauthors)